# CoFeS_2_@CoS_2_ Nanocubes Entangled with CNT for Efficient Bifunctional Performance for Oxygen Evolution and Oxygen Reduction Reactions

**DOI:** 10.3390/nano12060983

**Published:** 2022-03-16

**Authors:** Jaeeun Jeon, Kyoung Ryeol Park, Kang Min Kim, Daehyeon Ko, HyukSu Han, Nuri Oh, Sunghwan Yeo, Chisung Ahn, Sungwook Mhin

**Affiliations:** 1Korea Institute of Industrial Technology, 113-58, Siheung 15014, Korea; jaeeun00@kitech.re.kr; 2Korea Institute of Industrial Technology, 55, Ulsan 44413, Korea; krpark@kitech.re.kr; 3Korea Institute of Industrial Technology, 137-41, Gangneung 25440, Korea; kmkim@kitech.re.kr; 4Department of Advanced Materials Engineering, Kyonggi University, Suwon 16227, Korea; godh3134@naver.com; 5Department of Energy Engineering, Konkuk University, Seoul 05029, Korea; hhan@konkuk.ac.kr; 6Department of Advanced Materials Engineering, Hanyang University, Seoul 04763, Korea; irunho@hanyang.ac.kr; 7Korea Atomic Energy Research Institute, Daedeok-Daero 989-111, Deajeon 34057, Korea

**Keywords:** core–shell structure, sulfides, oxygen evolution reaction, oxygen reduction reaction, carbon nanotubes

## Abstract

Exploring bifunctional electrocatalysts to lower the activation energy barriers for sluggish electrochemical reactions for both the oxygen evolution reaction (OER) and oxygen reduction reaction (ORR) are of great importance in achieving lower energy consumption and higher conversion efficiency for future energy conversion and storage system. Despite the excellent performance of precious metal-based electrocatalysts for OER and ORR, their high cost and scarcity hamper their large-scale industrial application. As alternatives to precious metal-based electrocatalysts, the development of earth-abundant and efficient catalysts with excellent electrocatalytic performance in both the OER and the ORR is urgently required. Herein, we report a core–shell CoFeS_2_@CoS_2_ heterostructure entangled with carbon nanotubes as an efficient bifunctional electrocatalyst for both the OER and the ORR. The CoFeS_2_@CoS_2_ nanocubes entangled with carbon nanotubes show superior electrochemical performance for both the OER and the ORR: a potential of 1.5 V (vs. RHE) at a current density of 10 mA cm^−2^ for the OER in alkaline medium and an onset potential of 0.976 V for the ORR. This work suggests a processing methodology for the development of the core–shell heterostructures with enhanced bifunctional performance for both the OER and the ORR.

## 1. Introduction

The accelerated depletion of fossil fuels and accompanying environmental pollution have driven the development of advanced technologies for highly efficient energy conversion and storage systems, including fuel cells, metal–air batteries, and water electrolysis systems [1,2,3,4]. Particularly, recent technological advances in bifunctional electrocatalysts boosting both the oxygen evolution reaction (OER) and the oxygen reduction reaction (ORR) are of great importance, and have enabled the reduction of energy consumption and the enhancement of energy conversion efficiency for sustainable energy systems [5]. So far, precious-metal based electrocatalysts, including platinum, ruthenium oxide, and iridium oxide, for the OER and the ORR are employed as important components for industrial-level energy conversion and storage systems [6]. However, their scarcity on earth, poor durability, and high cost hamper the large-scale industrial application and thus, the exploration of nonprecious metal-based electrocatalysts with low cost, durability and high catalytic activity to both ORR and OER are significant [7]. Nonprecious transition metal chalcogenides consisting of transition metal atoms (i.e., Co, Ni, Fe) and chalcogen atoms (i.e., S, Se, Te) have attracted great attention due to their low cost, high electrical conductivity, and electrochemical durability for both the OER and the ORR [8,9,10]. Several design strategies for metal chalcogenides, including structure, phase, and defect engineering, have been suggested to increase the active site exposure and to reduce the energy barrier of catalytic reactions to enhance the performance of both the OER and the ORR [11]. Among them, the design of the core–shell structure is considered as one of the effective strategies, not only to increase the contact area at the interface of the heterostructure, but also to provide the synergetic electrochemical performance of the constituents. For example, J. Bai et al. reported that the core–shell Co_9_S_8_@MoS_2_ heterostructure shows outstanding OER and ORR performance for water splitting and Zn–air batteries, respectively [12]. To improve the electrocatalytic activity of the transition metal chalcogenides further for both the OER and the ORR, conductive supports, including graphene, reduced graphene oxide and graphene/carbon nanotubes, are considered to promote electron transfer during the OER and ORR [13,14,15,16,17,18]. It is reported that porous nitrogen-doped carbon as a conductive support for cobalt sulfide promotes high catalytic activity and durability for both the OER and the ORR [19]. In addition, M. Shen et al. reported that encapsulation of cobalt–iron double sulfides through carbon support using nitrogen-doped mesoporous graphitic carbon improves the electrocatalytic activity and durability for the ORR and the OER [20]. Therefore, there is value in developing the transition metal based core–shell heterostructures combined with conductive support for synergetic electrochemical performance for both the OER and the ORR.

Herein, we report a novel synthesis route to prepare core–shell CoFeS_2_@CoS_2_ nanocubes entangled with carbon nanotubes via a hydrothermal method, followed by post-annealing under a reducing atmosphere. CoFeS_2_@CoS_2_/CNTs show outstanding OER activity, with an overpotential of 269 mV at a current density of 10 mA cm^−2^, a Tafel slope of 19.68 mV dec^−1^, and good durability in alkaline medium. In addition, CoFeS_2_@CoS_2_/CNTs show excellent ORR performance with an onset potential of 0.976 V, close to that of Pt/C.

## 2. Experimental Procedures

### 2.1. Preparation of CoS_2_/CNTs and CoFeS_2_@CoS_2_/CNTs

The CoS_2_/CNTs and CoFeS_2_@CoS_2_/CNTs samples were synthesized via a two-step process. In the first step, 1.05 mmol CoCl_2_·6H_2_O, 1.05 mmol FeCl_2_·6H_2_O, and carbon nanotubes (CNT) were dissolved in 30 mL deionized water. After Co, Fe, and CNTs powder were completely dissolved, 5 mg of thioacetamide (TAA) was added to this solution and magnetically stirred for 30 min. The obtained mixture was transferred into a Teflon-lined stainless steel autoclave and the autoclave was tightly sealed and kept at 150 °C for 12 h to produce CoS_2_/CNTs. After cooling to 25 °C naturally, the product was collected by centrifugation, washed several times thoroughly with deionized water, and dried at 60 °C for 8 h. Subsequently, dried CoS_2_/CNTs powder was placed in a crucible and calcined in a tube furnace under Ar atmosphere at 350 °C for 1 h with a ramping rate of 5 °C min^−1^ to obtain CoFeS_2_@CoS_2_/CNTs.

### 2.2. Material Characterization

The surface morphology and element composition were examined with a field emission scanning electron microscope (FE-SEM, NNS-450, FEI). Transmission electron microscope (TEM) and high-resolution transmission electron microscope (HRTEM) images were obtained using a TALOS F200X instrument. The corresponding element mapping images were obtained with image correctors. The crystal structural characterization of CoS_2_/CNTs and CoFeS_2_@CoS_2_/CNTs particles was performed by X-ray diffraction (XRD, X’Pert-Pro MPD, PANalytical, Malvern, UK) using Cu Kα radiation at 40 kV and 30 mA, in the range from 10° to 80° with a scan rate of 0.02° per second. The chemical valence state for chemical oxidation analysis was determined by X-ray photoemission spectroscopy (XPS, ESCALAB 250Xi, Thermo Fisher Scientific, Waltham, MA, USA).

### 2.3. Electrochemical Characterization

Electrochemical measurements were conducted on an electrochemical workstation (model Autolab PGSTAT; Metrohm) in 0.1 M and 1.0 M KOH and evaluated using a standard three-electrode electrochemical cell with a rotating disk electrode (RDE). For the preparation of the working electrode, a sample was dispersed in a Nafion^®^ solution (10 μL, 5 wt%) of water and ethanol (volume ratio 3:1). Samples were tested in 0.1 M and 1 M KOH aqueous solution by using a glassy carbon (GC) electrode as the working electrode, with a diameter of 3 mm, which yielded an approximate catalyst loading of 0.253 mg cm^−2^. The working electrode was dried at room temperature prior to electrochemical measurements. Before testing the ORR, the electrolytes were generally saturated with O_2_, and the flow rate of oxygen was maintained during the whole test. The obtained potentials (vs. Ag/AgCl) were calibrated to the reversible hydrogen electrode (RHE) according to the Nernst equation (E_RHE_ = E_Ag/AgCl_ + 0.0591pH + 0.197). The polarization curves were measured by linear sweep voltammetry (LSV) at a scan rate of 5 Mv s^−1^. Electrochemical impedance spectra (EIS) were recorded under an AC voltage of 5 mV with frequencies in the range of 50 kHz to 0.1 Hz. Cyclic voltammetry (CV) was performed in the range of 1.32 to 1.48 V (vs. RHE) with scan rates of 20 mV, 40 mV, 60 mV, 80 mV, 100 mV, and 120 mV. The electrical double layer specific capacitance (C_dl_) of catalysts was determined from CV. Overall water splitting tests were performed in an electrode configuration with CoS_2_/CNTs and CoFeS_2_@CoS_2_/CNTs.

## 3. Results and Discussion

The XRD patterns of CoS_2_/CNTs and CoFeS_2_@CoS_2_/CNTs are presented in Figure 1a. CoS_2_ forms after hydrothermal processing. Subsequent post annealing under reducing conditions transforms CoS_2_ to mixed CoFeS_2_ and CoS_2_ phases_._ Further, the diffracted peaks at 26° indicate the presence of CNTs in both samples. It is noted that similar X-ray diffraction patterns were observed from pure CoFeS_2_@CoS_2_, except for the diffracted peak at 26° indicating CNTs, as shown in Appendix A. Electron micrographs of CoS_2_/CNTs show that nanoparticles with irregular shapes are entangled with CNTs, as shown in Figure 1b,c. After post annealing under reducing conditions, the irregularly shaped CoS_2_ particles transform to CoFeS_2_@CoS_2_ nanocubes. CoFeS_2_@CoS_2_ without CNTs also has a similar microstructure, as shown in Appendix A. TEM images and the corresponding EDS maps indicate the CoS_2_ nanoparticles with a lattice spacing of 1.62 Å of (200) and 2.47 of (210), which become entangled with CNTs after hydrothermal processing, as presented in Figure 1d–f. After post annealing, the CoS_2_ nanoparticles transform to the core(CoFeS_2_)–shell(CoS_2_) nanocubes entangled with CNTs, confirmed by the outer CoS_2_ and inner CoFeS_2_ with a lattice spacing of 2.47 Å of (210) and 1.95 Å of (100), respectively (Figure 1g–i). The Selected area diffraction (SAED) pattern of CoFeS_2_@CoS_2_ is shown in Appendix A. It is clear that the diffraction rings with d-spacings of 0.247 and 0.162 nm correspond to (210) and (200) of CoS_2_, respectively. In addition, the diffraction rings with d-spacing of 0.195 nm correspond to (100) of CoFeS_2_. It is noted that morphology of the CNTs remain similar after synthesis. 

To better understand the effect of post annealing under reducing conditions on the chemical nature of constituent ions, XPS analysis was performed, and results are shown in Figure 2. There are three major peaks of the XPS spectra, Co 2p (Figure 2a), Fe 2p (Figure 2b), and S 2p (Figure 2c), which describe the chemical nature of CoS_2_ and CoFeS_2_@CoS_2_. As shown in Figure 2a,b, the Co 2p states of the CoS_2_ are observed at 781.3 eV and 783.1 eV (Co 2p3/2), 797.5 eV and 801.7 eV (Co 2p1/2), and 786.3 eV and 805.1 eV (satellite peaks), while Co 2p states of CoFeS_2_@CoS_2_ are 778.4 eV and 780.2 eV (Co 2p3/2), 793.6 eV and 796.7 eV (Co 2p1/2), and 782.7 eV and 803.1 eV (satellite peaks). In addition, the Fe 2p states of CoS_2_ are observed at 711.7 eV and 716.5 eV with satellite peaks at 724.1 eV, while the Fe 2p states of CoFeS_2_@CoS_2_ are at 712.1 eV and 716.8 eV with satellite peaks at 724.5 eV. The S 2p states of CoS_2_ are observed at 163.8 eV (S 2p3/2), while the S 2p states of CoFeS_2_ are observed at 162.7 eV (S 2p3/2) and 164.2 eV (S 2p1/2), as shown in Figure 2c. The XPS results indicate that both CoS_2_ and CoFeS_2_ contain the oxidation states of Co^2+^ and Co^3+^, Fe^2+^ and Fe^3+^, and S_2_^2−^**.** It is noted that reducing conditions during post annealing of CoS_2_ stabilizes Fe^2+^ and thus, the peak shift of the Fe 2p state toward a lower binding energy (Figure 2b), which promotes the incorporation of the Fe^2+^ into the Co sites of CoS_2_. In addition, the peak shift of the Co 2p and S 2p states towards a lower binding energy indicates the modulated metal–sulfur bonding induced by Fe incorporation into the Co sites of CoS_2_, as shown in Figure 2a,c [21,22,23,24].

The LSV polarization curves of CoS_2_/CNTs and CoFeS_2_@CoS_2_/CNTs are shown Figure 3a. Compared to the overpotential of CoS_2_/CNTs and IrO_2_, CoFeS_2_@CoS_2_/CNTs show superior OER activity, with an overpotential of 269 mV and 320 mV at 10 mA cm^−2^ and at 100 mA cm^−2^, respectively. A recent DFT calculation study suggests that the electronic states of Co are tuned by synergetic interaction between sulfur and the adjacent iron, which can lower the overpotential for OER process [25]. In addition, compared to pure CoFeS_2_@CoS_2_, CoFeS_2_@CoS_2_/CNTs shows a lower overpotential, smaller Tafel plot, and resistance of charge transfer, in Appendix A, respectively. This indicates that the catalytic activity of the CoFeS_2_@CoS_2_ with CNTs is superior to that of CoFeS_2_@CoS_2_, which can also explain the important role of the CNTs in boosting the catalytic activity of the CoFeS_2_@CoS_2_ [26,27,28,29].

The Tafel slopes of the CoS_2_/CNTs and CoFeS_2_@CoS_2_/CNTs samples were also investigated to understand the reaction kinetics (Figure 3b). The CoFeS_2_@CoS_2_/CNTs exhibits a relatively lower Tafel slope of 19.68 mV dec^−1^, compared to that of CoS_2_/CNTs (24.06 mV) and IrO_2_ (42.28 mV dec^−1^), which indicates more the favorable OER kinetics of the CoFeS_2_@CoS_2_/CNTs. The CoFeS_2_@CoS_2_/CNTs show superior OER performance compared to the recently reported state-of-the-art OER electrocatalysts (Appendix A). In addition, the higher C_dl_ of the CoFeS_2_@CoS_2_/CNTs (43.67 mF cm^−2^) compared to that of CoS_2_/CNTs (16.21 mF cm^−2^) suggests that the improved OER activity of the CoFeS_2_@CoS_2_/CNTs can be attributed to the large electrochemically active surface area (ECSA), as shown in Figure 3c and Appendix A. For practical application, the long-term stability is also important factor to evaluate the electrochemical performance of the OER electrocatalysts. The electrocatalytic stability of the CoFeS_2_@CoS_2_/CNTs was evaluated using chronoamperometric measurements at an overpotential of 269 mV at 10 mA cm^−2^. After 12 h of the OER, the current density of the CoFeS_2_@CoS_2_/CNTs was decreased by less than 3% of its initial value (Figure 3d). During the long-term OER stability test, Co 2p (Figure 4a), Fe 2p (Figure 4b), and S 2p (Figure 4c) peaks of the XPS spectra were changed. It is noted that the initial j/j_0_ value exceeding 100% can be attributed to the excellent OER activity of the electrochemically active byproducts formed during the OER process, which is also reported in previous literature [30]. Moreover, a slight change in the valence states of the Co and Fe in CoFeS_2_@CoS_2_/CNTs was observed: both Co 2p and Fe 2p peaks are slightly shifted to higher binding energy due to the oxidation on the electrode surface, which suggests the tolerance of the optimized CoFeS_2_@CoS_2_/CNTs against corrosion under long-term oxidizing conditions. This can be attributed to the synergistic interaction of CoFeS_2_@CoS_2_ with CNTs, which improves the stability of CoFeS_2_@CoS_2_/CNTs [31,32].

Most importantly, the S 2p peak indicating metal–sulfur bonding at 163.7 and 163.8 eV disappeared, while a new S 2p peak indicating the sulfur–oxygen bonding at 168.5 eV appears, implying that the oxidation of sulfide occurs [33]. As shown in Appendix A, the oxidation of CoFeS_2_@CoS_2_ leads to the transformation to various chemical compounds, including FeOOH, CoO, and CoFe_2_O_4_. Notably, the formation of the FeOOH nanosheet and CoFe_2_O_4_ on the surface of CoFeS_2_@CoS_2_ can provide active sites to promote the OER process [34,35]. However, the formation of the less active CoO phase can lead to the gradual deterioration of CoFeS_2_@CoS_2_ during the OER process. 

The electrocatalytic activity of CoS_2_/CNTs and CoFeS_2_@CoS_2_/CNTs for ORR is analyzed by obtaining the LSV curves as shown in Figure 5a. CoFeS_2_@CoS_2_/CNTs exhibits ORR performance with an onset potential (E_onset_) of 0.976 V and a half-wave potential (E_1/2_) of 0.871 V (vs. RHE), which is superior to that of CoS_2_/CNTs. CoFeS_2_@CoS_2_/CNTs show a lower Tafel slope (24.43 mV dec^−1^) than CoS_2_/CNTs (78.31 mV dec^−1^) and Pt/C (25.73 mV dec^−1^), which indicates the superior catalytic activity of CoFeS_2_@CoS_2_/CNTs (Figure 5b). To further understand the kinetics of the ORR for CoFeS_2_@CoS_2_/CNTs, Koutecký–Levich (K-L) plots were evaluated as shown in Figure 5c. The average number of the transferred electrons per oxygen molecule was calculated as 4.3 in the potential range between 0.35 and 0.5 V using the K-L equation, which indicates that CoFeS_2_@CoS_2_/CNTs follows a four-electron pathway for the ORR. It is noted that the four-electron pathway for the ORR is preferred over the two-electron pathway for higher energy conversion efficiency as complete oxygen reduction occurs [29]. In addition, the polarization curve of CoFeS_2_@CoS_2_/CNTs exhibits negligible current loss after 2000 cycles at a scan rate of 10 mV s^−1^. (Figure 5d) After the ORR cycle test, the oxidation of CoFeS_2_@CoS_2_/CNTs was also observed, which transforms to various chemical compounds, including CoO and CoFe_2_O_4_, as shown in Appendix A. This observation is well-matched with the experimental results of previous studies, which reveals the transformation of transition metal-based sulfide into various oxides under ORR conditions [36,37,38]. Based on the electrochemical performance of CoS_2_/CNTs and CoFeS_2_@CoS_2_/CNTs, the potential difference (ΔE = E_j=10_ − E_1/2_) between E_j=10_ for the OER and E_1/2_ for the ORR of CoFeS_2_@CoS_2_/CNTs has a smaller ΔE (0.67 V) than CoS_2_/CNTs (ΔE = 0.82 V) as shown in Figure 5e, which suggests the outstanding bifunctional (OER/ORR) catalytic activity of CoFeS_2_@CoS_2_/CNTs. 

## 4. Conclusions

A core–shell CoFeS_2_@CoS_2_ heterostructure entangled with carbon nanotubes is prepared by a facile hydrothermal synthesis, followed by post heat treatment under a reducing atmosphere. The CoFeS_2_@CoS_2_/CNTs electrocatalyst shows highly efficient OER performance in alkaline solution, with an overpotential of 269 mV and yielding a current density of 10 mA cm^−2^ with a Tafel slope of 19.68 mV dec^−1^. In addition, the CoFeS_2_@CoS_2_/CNTs electrocatalyst has outstanding ORR performance, with an onset potential (E_onset_) of 0.976 V and a half-wave potential (E_1/2_) of 0.871 V (vs. RHE). Given the overpotentials for OER (E_j=10_) and ORR (E_1/2_), the potential difference (ΔE = E_j=10_ − E_1/2_) is only 0.67 V, which shows the excellent catalytic activity of CoFeS_2_@CoS_2_/CNTs for the both OER and the ORR as efficient bifunctional electrocatalysts.

## Figures and Tables

**Figure 1 nanomaterials-12-00983-f001:**
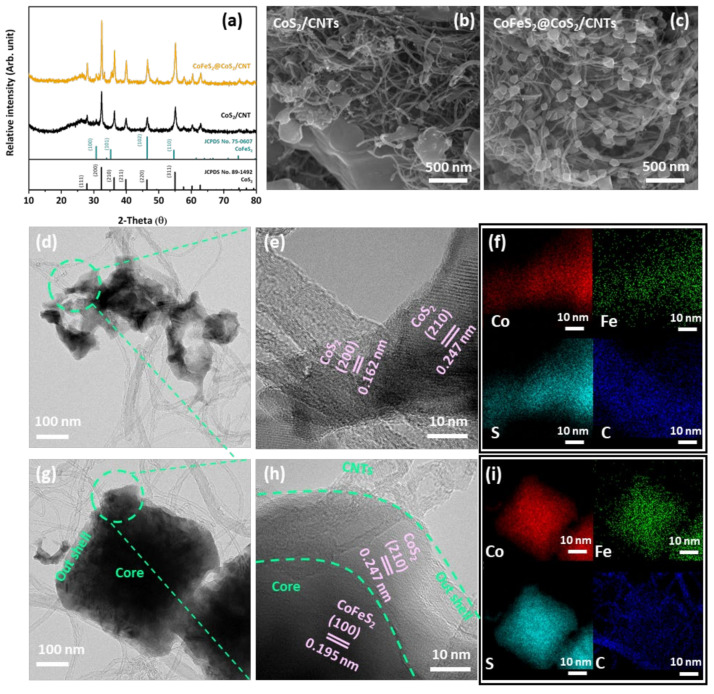
(**a**) XRD patterns of CoS_2_/CNTs and CoFeS_2_@CoS_2_/CNTs. FE-SEM image of (**b**) CoS_2_/CNT and (**c**) CoFeS_2_@CoS_2_/CNT. (**d**) TEM and (**e**) high-resolution TEM image of CoS_2_/CNTs with (**f**) elemental mapping images. (**g**) TEM and (**h**) high-resolution TEM images of CoFeS_2_@CoS_2_/CNTs with (**i**) HAADF-STEM image and elemental mapping images.

**Figure 2 nanomaterials-12-00983-f002:**
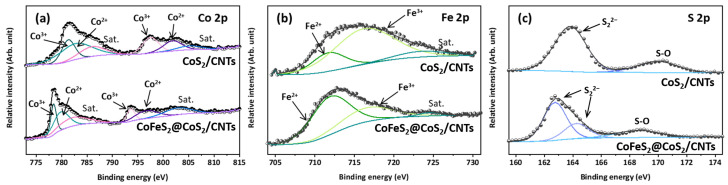
(**a**) Co 2p, (**b**) Fe 2p, and (**c**) S 2p spectra of CoS_2_/CNTs and CoFeS_2_@CoS_2_/CNTs.

**Figure 3 nanomaterials-12-00983-f003:**
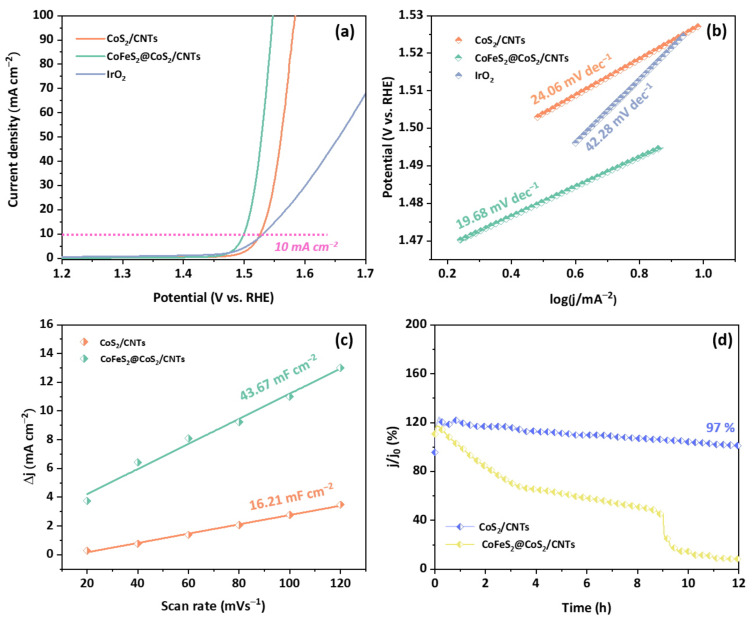
Electrocatalytic activity of CoS_2_/CNTs, CoFeS_2_@CoS_2_/CNTs, and IrO_2_ for the OER. (**a**) LSV polarization curves; (**b**) Tafel plots; (**c**) C_dl_; (**d**) long-term stability test at j = 10 mA cm^−2^.

**Figure 4 nanomaterials-12-00983-f004:**
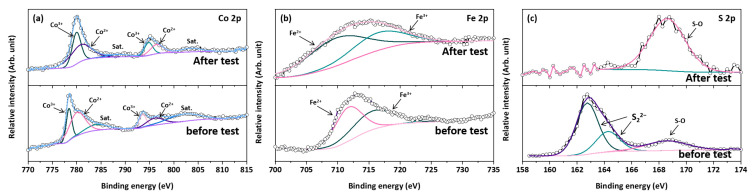
(**a**) Co 2p, (**b**) Fe 2p, and (**c**) S 2p spectra of CoFeS_2_@CoS_2_/CNTs before/after long-term stability test.

**Figure 5 nanomaterials-12-00983-f005:**
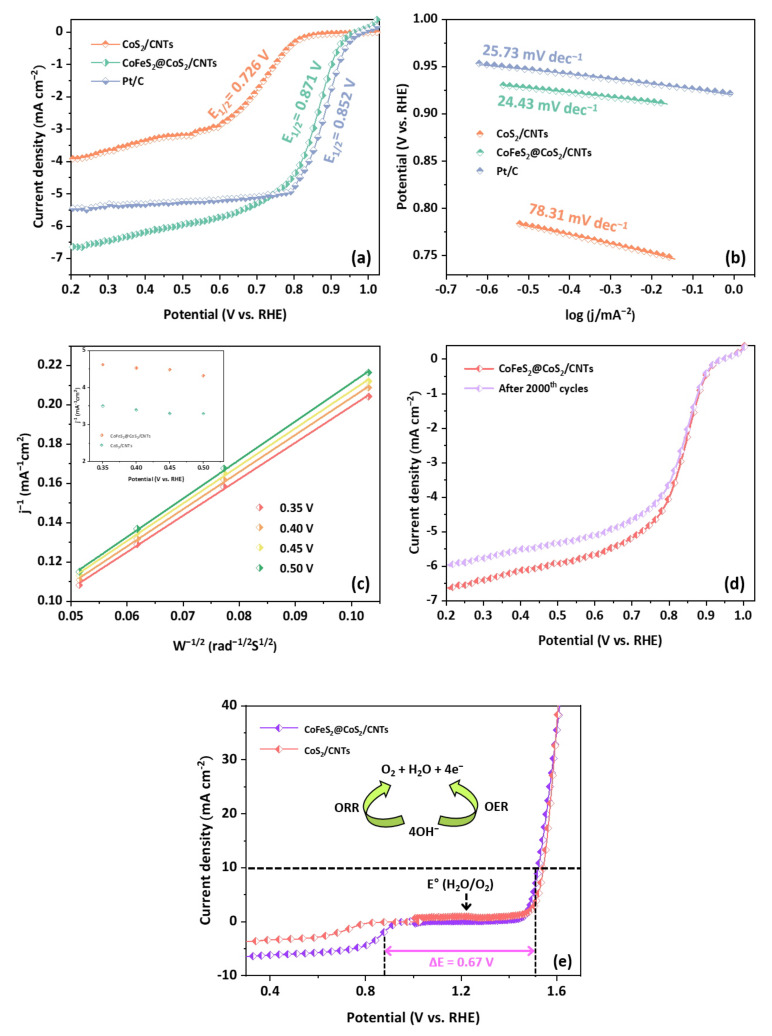
Electrocatalytic activity of CoS_2_/CNTs, CoFeS_2_@CoS_2_/CNTs, and Pt/C for the ORR: (**a**) LSV polarization curves; (**b**) Tafel plots; (**c**) K-L plot; (**d**) LSV polarization curves after 1st and 2000th cycles; (**e**) overall polarization curves of the catalysts within the ORR and OER potential.

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
