# Peer review of "CoFeS2@CoS2 Nanocubes Entangled with CNT for Efficient Bifunctional Performance for Oxygen Evolution and Oxygen Reduction Reactions"

_nanomaterials, 2022, doi:10.3390/nano12060983_

Round 1
Reviewer 1 Report
The authors report on the synthesis of core-shell CoFeS2@CoS2 nanocubes combined with carbon nanotubes, applied as bifunctional electrocatalyst for both oxygen evolution and oxygen reduction reactions. The materials were prepared by hydrothermal treatment and post-thermal reduction to obtain the core-shell structure, while evaluation was carried out by electrochemical measurements. Overall, the work is well designed and very well-presented reporting interesting results on transition metal chalcogenide based electrocatalysts in combination with graphene nanomaterials for OER and ORR water splitting. The following points should be though considered:
1) The beneficial role of CNTs as conducting support to the CoFeS2@CoS2 nanoparticles should be shown by some comparative tests.
2) How is CoS2 discriminated from CoFeS2 by diffraction in the TEM images?
3) Performance comparison with similar systems reported in the literature, as those described in the introduction should be briefly made.
4) There are some typos in the manuscript that should be corrected e.g. “nonocubes” in the title, “wished several times” in Experimental procedures.
Reviewer 2 Report
This manuscript contains certain novelty, and is well organized. I recommend its acceptance by Nanomaterials after addressing the following issues.
- There are many typos throughout the manuscript. For example, in Title, “nonocubes” should be “nanocubes”, and in Experimental section, Line 8 of the 1st paragraph, “wished” should be “washed”, etc. Such mistakes should be carefully checked and corrected.
- Interfacial effects, for example charge transfer and valence state regulation, between CoFeS2 and CoS2, as well as sulfides and CNTs, should be discussed in detail. Suggested references: (1) Chem. Eng. J. 2021, 424, 130444; (2) J. Alloys Compd. 2020, 827, 154163; (3) J. Energy Chem. 2021, 55, 92-101; etc.
- In Fig. S2, it says the oxidation products of CoFeS2@CoS2 after 12-h OER include FeOOH, CoO and CoFe2O4. However, only the HRTEM images are not enough. XRD and XPS should also be provided to support this conclusion. Besides, considering that FeOOH and CoFe2O4 displayed high OER activity in many previous reports, is it reasonable that the j/j0 faded to <10% after 12 hour?
- Characterization results of CoS2/CNTs after 12-h OER, and CoFeS2@CoS2/CNTs after 2000 cycles of ORR should be shown as well to confirm their stability.
Reviewer 3 Report
Current work reports CoFeS2@CoS2 nanocubes entangled with CNT for efficient bi-functional performance to oxygen evolution and oxygen reduction reaction, which is proposed to be excellent catalytic performance for OER and ORR.
Despite the excellent electrochemical performance, particularly, the OER performance, the paper is riddled with inaccuracies. Before considering the manuscript for publication, authors are advised to make major revisions.
- I guess the term‘‘ nonocubes’’ must be ‘‘nanocubes’’ in the title, such a mistake in title indicates that authors didn’t revise the manuscript well before submission. Similarly, the following lines has been repeated, ‘‘There are three major Co 2p (Fig. 2a), Fe 2p (Fig. 2b) and S 2p (Fig. 2c) peaks of the XPS spectra to describe chemical nature of the CoS2 and CoFeS2@CoS2. There are three major Co 2p (Fig. 2a), Fe 2p (Fig. 2b) and S 2p (Fig. 2c) peaks of the XPS spectra to describe chemical nature for the CoS2 and CoFeS2@CoS2’’.
- SEM images show bulky particles attached To CNTs, however, based on the CNTs diameter, it is unlikely to explain the interaction between CoFeS2@CoS2 NPS and CNTs. Authors are recommended to comment on it.
- XRD spectra of pure CoFeS2@CoS2 without CNTs should provided for the convincing comparison.
- TEM and HR-TEM in (Figure 1d, e and g, h) are not consistent. Authors are suggested to provide more accurate HR-TEM images.
- XPS results are not explained well with reference to synergistic interaction of catalyst NPs with support material.
- Authors mentioned synergetic interaction of transition metals with catalyst NPS for improved electrochemical performance of OER and ORR. As mentioned above, what is the basis of interaction of CNTs with CoFeS2@CoS2. Following recent references might be helpful regarding the phenomena, (S. Zaman et al. Angew. Chem. Int. Ed. 2021, 60, 17832-17852, A. I. Douka et al. Adv Mater 2020, 32, e2002170, Zaman et al. Angew. Chem. Int. Ed. 2021, 61, e202115835, S. Zaman et al. Sci. Bull. 2021, 66. 2201).
- Authors should consider the j/jo value, which more than 100% in figure 5d.
- It is recommended that authors thoroughly review their article for grammatical and typographical errors.

Round 2
Reviewer 2 Report
Since the authors have addressed all my comments, I think it can be accepted by Nanomaterials at current version.
* Ref. 35: The journal title is wrong. It should be "Chem. Eng. J." not "J. Energy Chem.".
Reviewer 3 Report
Accepted